# Strengths of Exaggerated Tsunami-Originated Placenames: Disaster Subculture in Sanriku Coast, Japan

**Yuzuru Isoda** [1,*], **Akio Muranaka** [2], **Go Tanibata** [3], **Kazumasa Hanaoka** [2], **Junzo Ohmura** [4] **and Akihiro Tsukamoto** [5]

1   Graduate School of Science, Tohoku University, Sendai 980-8578, Japan
2   Department of Geography, College of Letters, Ritsumeikan University, Kyoto 603-8577, Japan;
    muranaka@fc.ritsumei.ac.jp (A.M.); kht27176@fc.ritsumei.ac.jp (K.H.)
3   Faculty of Law, Miyazaki Sangyo-keiei University, Miyazaki 880-0931, Japan;
    tanibata@mail.miyasankei-u.ac.jp
4   Earthquake Research Institute, the University of Tokyo, Tokyo 113-0032, Japan; ohmura@eri.u-tokyo.ac.jp
5   Faculty of Integrated Arts and Sciences, Tokushima University, Tokushima 770-8501, Japan;
    tsukamoto.akihiro@tokushima-u.ac.jp
*   Correspondence: isoda@tohoku.ac.jp

**Abstract:** Disaster-originated placename is a kind of disaster subculture that is used for a practical purpose of identifying a location while reminding the past disaster experience. They are expected to transmit the risks and knowledge of high-risk low-frequency natural hazards, surviving over time and generations. This paper compares the perceptions to tsunami-originated placenames in local communities having realistic and exaggerated origins in Sanriku Coast, Japan. The reality of tsunami-originated placenames is first assessed by comparing the tsunami run-ups indicated in the origins and that of the tsunami in the Great East Japan Earthquake 2011 using GIS and digital elevation model. Considerable proportions of placenames had exaggerated origins, but the group interviews to local communities revealed that origins indicating unrealistic tsunami run-ups were more believed than that of the more realistic ones. We discuss that accurate hazard information will be discredited if it contradicts to the people's everyday life and the desire for safety, and even imprecise and ambiguous information can survive if it is embedded to a system of local knowledge that consistently explains the various facts in a local area that requires explanation.

**Keywords:** disaster subculture; tsunami; placenames; indigenous knowledge; risk perception; tsunami run-up; digital elevation model; GIS

---

## 1. Introduction

There has been a revival in the field of toponymy, or a study of placenames, once a largely discredited field within the discipline of geography [1]. The revival is on one hand for the practical purpose of easy translation of placenames to geographic coordinates, but also to a new strand of toponymy, which may be called 'thematic toponymy'. Thematic toponymy scrutinizes a subset of placenames along a certain theme, breaking away from the encyclopedic nature of the traditional toponymy, to reveal spatial distribution, frequencies, or to restore the past geographies of a theme. The focus shifted from collecting, classifying and seeking origins for names, to shedding light to larger questions in social science, humanities, and natural science: the goals that the pioneers of toponymy have envisaged [2]. This new strand has been made possible partly to the collection of placenames by the earlier toponymists, the national databases and gazetteers, and the widespread use of GIS.

Biogeographers and biologists are the main users of the thematic toponymy, observing the past distributions of, for example, martens [3], ravens [4], eagles [5], wildlife species [6], tree species [7], and so forth from the present or the past placenames. Time-series analysis of landscape is also done through thematic toponymy. Sousa et al. use placenames in a series of historic maps since the 17th century to observe the land use changes [8], and the decrease in wetlands associated with the climate change [9]. Frajer and Feidor use a similar approach in identifying transformations of water bodies through modernization [10].

Toponymy has also gone through a 'critical turn' where placenames are seen as manifesting power-relations of the area and the act of naming as exercising political power over space [11–13]. Critical toponymy focuses on language and is mostly critical-interpretative but its integration with modern empirical methods such as GIS and statistical analysis has been advocated [14]. The integration enabled examination of spatial distributions and changes in the local power-relations [14–17], in a similar way historical geographers integrated GIS to their toponymic research [18–20].

Thematic toponymy, therefore, is not a study of the placenames per se, but rather uses placenames as a source of information to reveal a certain theme, whether it be in natural science, social science, or humanities. Other examples include the use of street names as indicators to local social and cultural characteristics [21], to use field names for management and planning of landscapes [22], and to use toponym databases to reveal different conceptualization of landscapes across different groups and cultures [23]. Uncertainties and ambiguities present in the placename information can be costly to rectify, as the very use of placename information is often because there is no other source, but is done through fieldwork [22,24] and cross-referencing to other available documents [5,9,10]. If the number of placenames for the analysis is large, the problem may be solved by applying statistical analyses [14,15].

In Japan, placenames have attracted attention in popular books and media as indications to disaster propensity and the past disaster experience, especially after the Great East Japan Earthquake 2011 [25]. Scientific examination to whether placenames indicate the past or future disasters are limited but a close match between the historical placename and a flooding disaster [26], between the placename suffix and tsunami risk [27], and between the bus stop name and seismic hazard [28], are found. The causal relationship behind these matches is because many places are named after local topography [26–28]. Hazard risk would then be better predicted by geomorphological classification, but the placenames are more informative because they are used by local people in everyday life, and thus more accessible, and they reveal the original topography sometimes obscured by recent developments.

Incorporation of indigenous knowledge, in contrast to scientific knowledge, to disaster management has been advocated by social scientists [29]. The exploration into and utilization of indigenous knowledge is encouraged in the Hyogo Framework for Action 2005–2015 [30], and is inherited to the Sendai Framework for Disaster Risk Reduction 2015–2030 [31]. More recently, Japanese tsunami engineers are focusing on the media by which the indigenous knowledge on tsunami is transmitted, such as stone monument, folklore, placename, and ruins, calling them collectively 'tsunami traditional knowledge media', as means to combat degradation of the awareness and the knowledge of the high-risk low-frequency hazards [32].

A concept 'disaster subculture' has been around since the 1960s among social scientists in disaster research, as "the assemblage of cultural practices that over time emerges in response to recurring disasters" ([33], p. 862). A celebrated example comes from Simeulue Island in Aceh, Indonesia, where song and stories recounting the 1907 tsunami had saved thousands of lives from the 2004 and 2005 tsunamis [34] (some more examples found in [35]). The strength of disaster subculture in disaster mitigation comes from the following ingredients: (1) Disaster experience is woven into a culture, in songs, stories, ceremonies, etc. (creation); (2) the culture embedded in everyday social life enables it to survive during the normal periods (transmission); and (3) people act based on the culture before, during and after a disaster (action). Disaster subculture is not without its flaws, however, Donovan reports a case in which local communities believing in local legend refused to follow the government warnings [36], and Shannon reports a case in which local communities evacuating after a

false premonition [37]. Both authors, however, says studying indigenous knowledge to reveal the way in which local people understand and act upon natural hazards is necessary for designing effective disaster risk reduction measures.

A disaster-originated placename is a typical and suitable example of disaster subculture. A place is named after a disastrous event, the name is used for the practical purpose of identifying a location while reminding the previous disasters, and used as a reference in making a spatial decision such as housing site selection, evacuation or housing reconstruction. They are also place-specific disaster subculture. Folklore and myths often propagate from elsewhere and thus may not be relevant in a particular place, but a disaster-originated placename manifest that a disaster occurred on that specific location.

For a disaster-originated placename to be an effective disaster subculture, it has to meet several conditions. First, the disaster experience at the location should be true; otherwise, people would be misguided or would not treat the origin seriously if the name appears to be unreliable. Second, not only the name but also the origin of a placename has to be transmitted correctly and remembered. Finally, the origin has to be believed for the people to base their actions.

This paper is thematic toponymy of tsunami-originated placenames in Sanriku Coast in Iwate Prefecture, Japan. The region has a wealth of tsunami-originated placenames, but their roles as disaster subculture are not assessed. The purpose of the study is to reveal how people understand tsunami-originated placenames and whether the placename is contributing to maintaining awareness and in making actions, in the face of uncertainty and ambiguity in the origins. For these purposes, we first assess the reality of the origins to tsunami-originated placenames. Our premise is that for a disaster-originated placename to function as disaster subculture, the origin has to be true. Second, we compare the perceptions to tsunami-originated placename in the local communities having more and less realistic placename origins, to understand how people deal with uncertainty and ambiguity in the origins. Finally, the possible roles of tsunami-originated placenames in disaster mitigation and disaster risk communication are discussed.

## 2. Materials and Methods

We use a subset of tsunami-originated placenames collected by Soshin Yamana in 1896 [38] that are confirmed to exist with their origin by Muranaka et al. [39], and other tsunami-originated placenames found during the fieldwork for the same study. Sanriku Coast, our study area, has repeatedly experienced tsunami disasters including the Great East Japan Earthquake and Tsunami 2011 (Tsunami 2011 hereafter), as the coast face the subduction zone along the Japan Trench. The southern part of Sanriku Coast is a ria coast, where valleys are submerging into the sea and its dendritic coastline amplifies the wave height of the tsunamis, whereas the northern part of the coast is rising and forming coastal terraces [40]. The previous tsunami disasters include Jogan Tsunami 869, Keicho Tsunami 1611, Enpo Tsunami 1677, Kansei Tsunami 1793, and Ansei Tsunami 1856, Meiji Tsunami 1896, Showa Tsunami 1933, and Chilean Tsunami 1960. Geological studies show that Jogan Tsunami 869 and Kecho Tsunami 1611 are similar in magnitude to the Tsunami 2011 from the corresponding tsunami deposits found in the Sendai Plain, in the south of Sanriku Coast [41].

It was immediately after Meiji Tsunami 1896 when Soshin Yamana, an entrepreneur from Tono, Iwate Prefecture, collected tsunami-originated placenames alongside his investigation on the damages in the villages along the Sanriku Coast. Yamana collected 40 placenames, not in any way a comprehensive collection, and Muranaka et al. identified exact locations of 21 places of which 11 were found to have their origins surviving among the local people [39]. These 11 tsunami-originated placenames and additional 14 place names that we confirmed their tsunami origin surviving are examined in this study (Figure 1).

Sanriku coast, having repeated tsunami disasters, is also home to other disaster subculture. *Tsunami-tendenko* tradition, popularized by Yamashita [42], is advice from forebears of the region that evacuation to higher grounds is an utmost priority when a tsunami strikes, with the term *tendenko* meaning individually, separately, and prioritizing saving owns life. Many villages have one or more

stone monuments commemorating the previous tsunamis, and the one in Aneyoshi settlement in Miyako City built after the Showa Tsunami 1933 stating "do not build houses lower than here" succeeded in keeping houses away from the lowland [43]. School children were taught of tsunami song typically with a lyric saying that there would be a tsunami after a large earthquake. Sato et al. find that the existence of stone monuments had a statistically significant effect in lowering the local death rates from the Tsunami 2011 [32]. The same study finds that tsunami-originated placenames had the reverse significant effect, but we believe the coverage of the tsunami-originated placename in that study is very limited, not including the very local tsunami-originated placenames examined in this study.

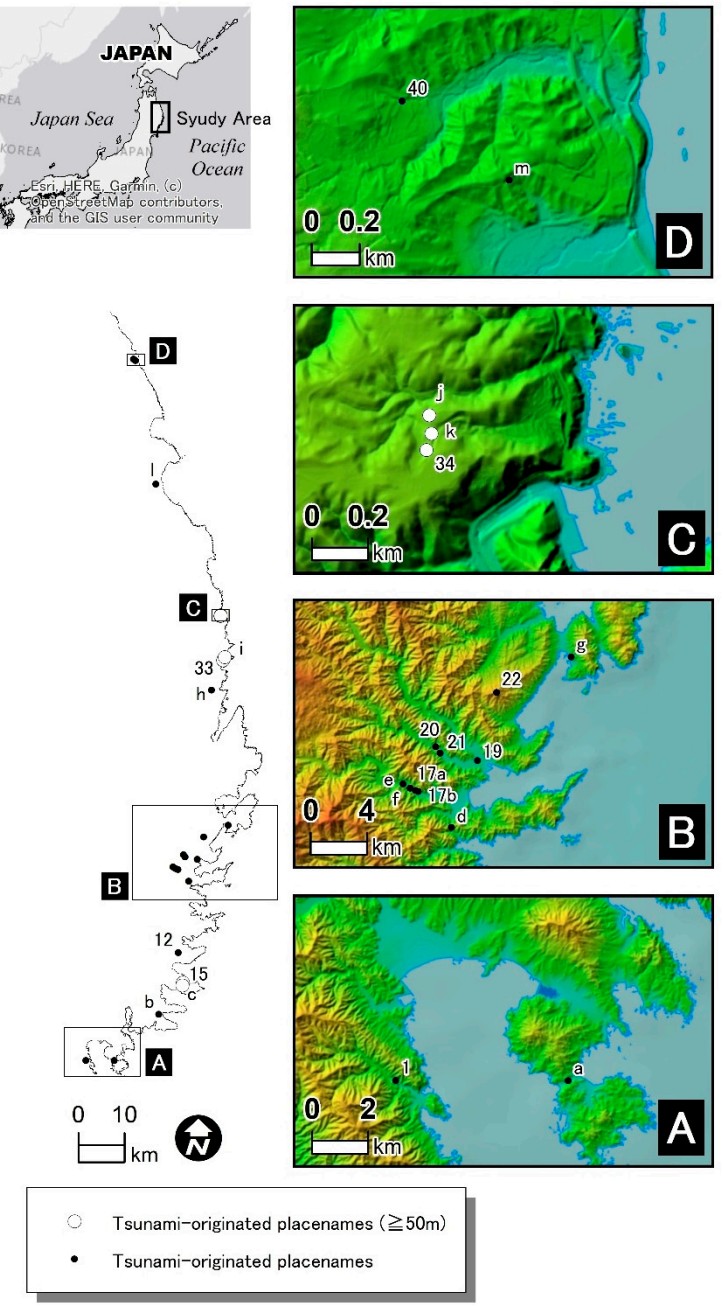

**Figure 1.** Map of tsunami-originated placenames. IDs in the map correspond to those in the list of placenames in Table 1. Empty dots are placenames with altitudes above 50 m.

**Table 1.** List of tsunami-originated placenames. IDs 1~40 are placenames in [38] and IDs a~m are those identified during fieldwork for [39]. Longitudes and latitudes are on the WGS 1984 datum.

| ID | Placename | Meaning | Feature | Longitude | Latitude |
|----|-----------|---------|---------|-----------|----------|
| 1 | Same-ga-fuchi | Shark Abyss | stream | E 141°37′ 17.60″ | N 38°58′ 1.99″ |
| 12 | Kuwa-dai | Hoe Plateau | mountain | E 141°51′ 20.90″ | N 39°10′ 33.39″ |
| 15 | Obune-zawa | Large Boat Stream | stream | E 141°51′ 57.81″ | N 39°07′ 1.53″ |
| 17a | Tagou | Octopus | settlement | E 141°51′ 17.69″ | N 39°20′ 20.38″ |
| 17b | Karagei | Small Ray Fish | house name | E 141°51′ 26.57″ | N 39°20′ 18.11″ |
| 19 | Tsuchi | Wooden Hammer | settlement | E 141°54′ 23.64″ | N 39°21′ 27.82″ |
| 20 | Sanmai-do | Three Doors | settlement | E 141°52′ 19.38″ | N 39°22′ 1.36″ |
| 21 | Usu-zawa | Mill Stream | settlement | E 141°52′ 32.93″ | N 39°21′ 46.40″ |
| 22 | Kujira-yama | Whale Mountain | mountain | E 141°55′ 22.86″ | N 39°24′ 4.98″ |
| 33 | Koe-ta | Crossed Over | settlement | E 141°58′ 37.90″ | N 39°44′ 45.35″ |
| 34 | Hachi-mori | Bowl Filled | dry field | E 141°58′ 21.03″ | N 39°49′ 58.14″ |
| 40 | Mekka-zawa | Seaweed Stream | stream | E 141°45′ 31.54″ | N 40°20′ 12.49″ |
| a | Atta | Met-Together | house name | E 141°41′ 32.68″ | N 38°57′ 59.89″ |
| b | Mizu-ai-toge | Water Meeting Pass | mountain pass | E 141°48′ 19.78″ | N 39°03′ 22.05″ |
| c | Tsuribachi-nagare | Hanging Bucket Pass | mountain pass | E 141°51′ 53.45″ | N 39°06′ 41.87″ |
| d | Koi-no-toge | Love Pass | mountain pass | E 141°53′ 3.32″ | N 39°18′ 54.24″ |
| e | Nagamochi | Portable Wardrobe | settlement | E 141°50′ 41.61″ | N 39°20′ 35.70″ |
| f | Keibai | Ravine Plum | settlement | E 141°51′ 2.32″ | N 39°20′ 25.77″ |
| g | Aburakko-zawa | Fat Greenling Stream | stream | E 141°59′ 6.94″ | N 39°25′ 24.34″ |
| h | Yagi-fune | Burnt Ship | location | E 141°56′ 50.40″ | N 39°41′ 18.62″ |
| i | Matsu-nagane | Pine Ridge | mountain ridge | E 141°58′ 54.95″ | N 39°45′ 7.74″ |
| j | Bozu-bata | Monk Field | dry field | E 141°58′ 21.53″ | N 39°50′ 2.18″ |
| k | Wano | Bowl Field | dry field | E 141°58′ 21.81″ | N 39°50′ 0.02″ |
| l | Aburakko-gappa | Fat Greenling Abyss | stream | E 141°48′ 45.24″ | N 40°05′ 30.46″ |
| m | Tago-ori | Octopus Washed-up | forest | E 141°45′ 50.34″ | N 40°20′ 1.65″ |

Our tsunami-originated placenames for the analysis are listed in Table 1. The meaning of the placenames can be categorized into two. The majority of placenames are the names of objects delivered by tsunamis, such as shark, octopus, large boat, etc. Association of the names of these objects to tsunamis is not obvious unless the origins were told, but names of objects related to the sea in inland areas may give rise to questions as to how the name came into place. All but one in the other category is about how waves behaved. Atta and Mizuai both describing the tsunami waves from two different sides 'met together', and Koe-no-toge and Koeta describing that the waves 'crossed over'. The only exception, Yagi-fune, describes the event after the tsunami of 'burning a boat' delivered from a tsunami after finding no other way to dispose of the wreckage.

Most named places are very local, such as a location in a stream, a mountain pass, a group of housing in a village, a patch of forest, a group of agricultural field parcels, and a house name. House name is used over generations and local people use house name to refer to the household or to the place of the house. These placenames are often used only by the villagers to identify the location within that village and hardly appear on maps. The locational specificity of these placenames allows us to find locations where the previous tsunami have reached. The tsunami-originated placename is a valuable source for past tsunami reach since there is no scientific method to identify past tsunami run-ups. This serves the basis for our first objective.

The reality of a tsunami-originated placename is assessed by comparing the tsunami reach indicated by the placename to that of Tsunami 2011. The Tsunami 2011 is just one specific tsunami and the comparison would not separate the origins to factual truth and false, but the tsunami is the one we have the most precise information and the comparison would be suitable for the purpose of gauging the reality of the origins.

The location of a named place is overlaid in GIS to a digital elevation model of approximately 10 m in spatial resolution from Geospatial Information Authority of Japan (GSI) to obtain the altitude, the distance from the sea along the drainage system, and the drainage basin it belongs. The comparisons

to Tsunami 2011 are done in two ways. The first comparison pairs the placenames to the furthermost reach of the Tsunami 2011 in the same drainage basin, recorded in the Haraguchi & Iwamatsu map of tsunami inundation and run-ups [44]. The differences in altitudes and the distance from the sea of the pair measures the reality of the placenames. The other comparison examines the statistical properties of the Tsunami 2011 inundation and run-up locations collected by TTJS group [45]. The TTJS group collected tsunami inundation and run-ups in various locations and often identified the highest tsunami reach in each drainage basin. The data is used to estimate means and standard deviations of altitudes and distances reached by the Tsunami 2011 based on an empirical relationship suggested by Kayane et al. [46]. A probability that Tsunami 2011 reaches the altitude and distance indicated by a placename is derived, to use for another measure of the reality of the placenames.

The location of the placename has to be specific to do the above comparisons, so the four placenames are omitted from the analysis: Kuwa-dai and Kujira-yama, both mountains, having the origin that a hoe and a whale, respectively, are delivered to the top or to the foot of the mountains; Tsuchi which is now part of the name for the whole municipality; and Aburakko-zawa which is the name of the entire stream. The remaining 21 places with tsunami origin will be analyzed.

Some communities had multiple tsunami-originated placenames. We conducted group interviews to two of such communities; Sakihama village in Ofunato City which have placename origins indicating tsunami run-up exceeding 80 m, and Unosumai Town in Kamaishi City which has a series of placenames indicating more realistic tsunami reaches. In Sakihama, we obtained cooperation from the chair of the Sakihama Public Interest Body, which acts as a neighborhood association in the village. In Unosumai Town, we obtained cooperation from the chair of the Kawame Neighborhood Association to gather interested residents from Kawame, where the placenames are located, and the rest of Unosumai Town.

We have chosen to do group interviews [47] because disaster subculture is the knowledge and practices of a community rather than that of an individual, and we were not expecting for any confidential response. Individuals can have different ideas and opinions, but we listened and waited until the discussion converges to what can be regarded as a community norm. If the number of attendants to group interview session were large, we split the attendants into groups of up to four persons so that everyone would be heard, and all the authors of this paper attended the sessions to facilitate discussions in each group. The interviews were recorded with consent, notes were taken and the recordings were replayed afterward to confirm and supplement the notes.

In a group interview session, we started from our presentation on various tsunami-originated placenames in Sanriku Coast to explain our interests, and then the followings were asked to be discussed among the participants: Is there additional or alternative information on the placename origin? From who and in what occasion did you learn the placename? On what occasions do you use the tsunami-originated placenames? Are there events or occasions that you talk about stories of the local area? Do you or on which occasions do you tell these origins to children or to grandchildren? Do you believe the origins to be true? The group interviews serve the basis for the second objective of this study, in revealing the perception of local communities to the less realistic and the more realistic origins of the tsunami-originated placenames.

Additionally, paroles from multiple individuals during fieldwork for Muranaka et al. study [39] and during the preparation for the above group interviews will also be used in the analysis. In all group and individual interviews, we explained our research purpose and gained consent to publicize the knowledge and opinions under the condition that personal identity will not be revealed.

## 3. Results

### 3.1. The Reality of Tsunami-Originated Placenames

To assess the reality of tsunami-originated placenames, we compare the tsunami reach indicated by the placenames and the tsunami inundation and run-ups recorded for Tsunami 2011 (Figure 2).

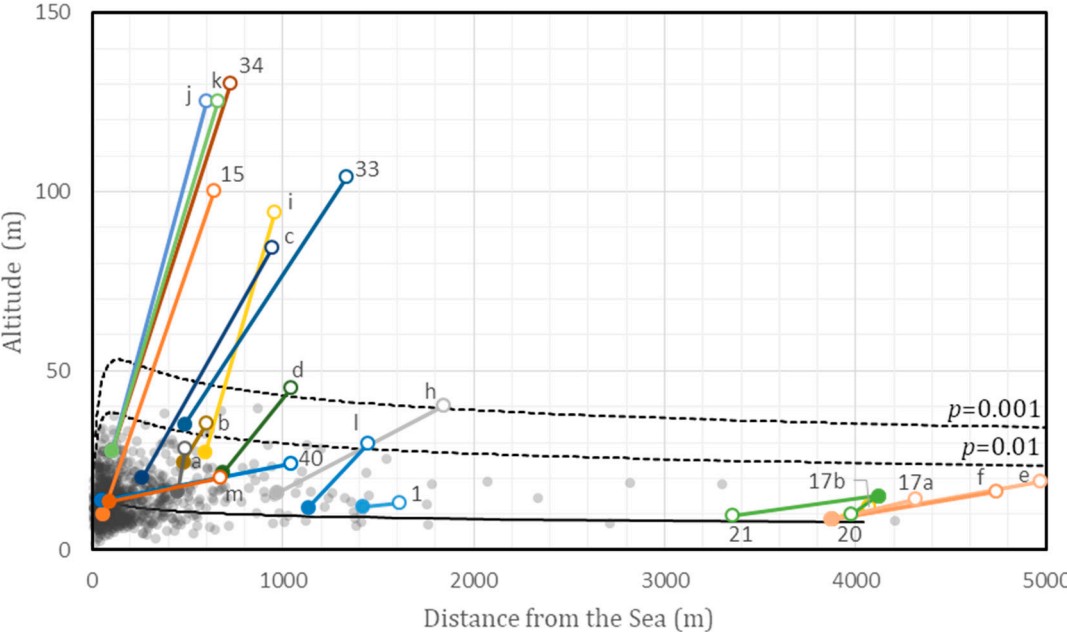

**Figure 2.** Paired tsunami reach and one-sided prediction intervals of reach of the Tsunami 2011. Tsunami reaches are indicated by tsunami-originated placenames (empty dots) and those of the Great East Japan Earthquake & Tsunami 2011 (filled dots). Distance from the sea is measured along the drainage of the basin. The tsunami reach indicated by a placename is matched to that of Tsunami 2011 in the same drainage basin in the Haraguchi & Iwamatsu map [44]. IDs in the graph correspond to those listed in Table 1. The dashed lines are 99% and 99.9% one-sided prediction interval of the highest tsunami reach of the Tsunami 2011, for drainage basins in Iwate Prefecture (shown with semi-transparent dots) in the TTJS data [45], and the solid line is the mean. The values for each placename is given in the Appendix A (Table A1) and the details of the prediction in Appendix B.

The difference in the matched pair in Figure 2 roughly gauges the reality of the tsunami reach indicated by the placenames. It shows that Tsunami 2011 did not reach the tsunami-originated places, except for the two: Usu-zawa and Sanmai-do, both of which are along the Kozuchi River in Otsuchi Town where the tsunami reached to an altitude of 15 m at a distance of 4119 m from the sea. There is a group of placenames in the upper left of the figure indicating heights exceeding 80 m, showing large discrepancies to the Tsunami 2011 reaches.

For the Tsunami 2011 in general, the highest tsunami height of 40.1 m is recorded in Ryori Bay in Ofunato City [46]. The highest run-up is at Aneyoshi in Miyako City, to an altitude of 39.7 m at the distance of 598 m from the sea [46,48], and the furthermost reach is along the Kyu-Kitakami River, at the distance of 33.6 km to an altitude of 7 m [46]. In Settai River in Tarou, Miyako City, the tsunami ran up to an altitude of 25.3 m at a distance of 2.5 km [46]. Mori et al. note that the local run-up heights differed considerably between neighboring locations, affected by sea walls, topography, debris, etc. [48].

The dashed lines in Figure 2 indicate one-sided prediction interval of Tsunami 2011 height and distance, from regression for the highest tsunami reaches in 1005 drainage basins in Iwate coast. The detail of the regression analysis is given in Appendix B. Out of 21 tsunami-originated placenames, nine placenames fall outside 99.9% interval, two fall between the upper bounds of 99.9% and 99%, and the remaining 10 fall within 99% interval.

There are three clusters of places falling outside 99.9% interval; Obune-zawa and Tsuribachi-nagare in Sakihama, Ofunato City; Koeta and Matsu-nagane in Koeta, Miyako City; and Hachi-mori, Wano, and Bozu-bata in Moshi, Iwaizumi Town. Places falling within 99% interval are also clustered into three:

Sanmai-do and Usu-zawa in Otsuchi Town where the Tsunami 2011 actually arrived; Mekka-zawa and Tago-ori in Hirono Town; and Tagou, Karagei, Keibai and Nagamochi in Unosumai Town.

*3.2. Perceptions of the Tsunami-Originated Placenames*

3.2.1. Sakihama, Ofunato City

Sakihama in Okirai, Ofunato City is a fishing village. The settlement lies in a small valley extending between the Sakihama Fish Port and the altitudes of about 70 m (Figure 3). There had been 978 persons in 532 households in 2010 Census. The 2011 Tsunami washed 48 houses situated below 10 m killing 10 persons, but the majority of houses were not affected. Most villagers are engaged in fishing and aquaculture but also in community forestry in which trees for timber were planted on previous common forests for firewood.

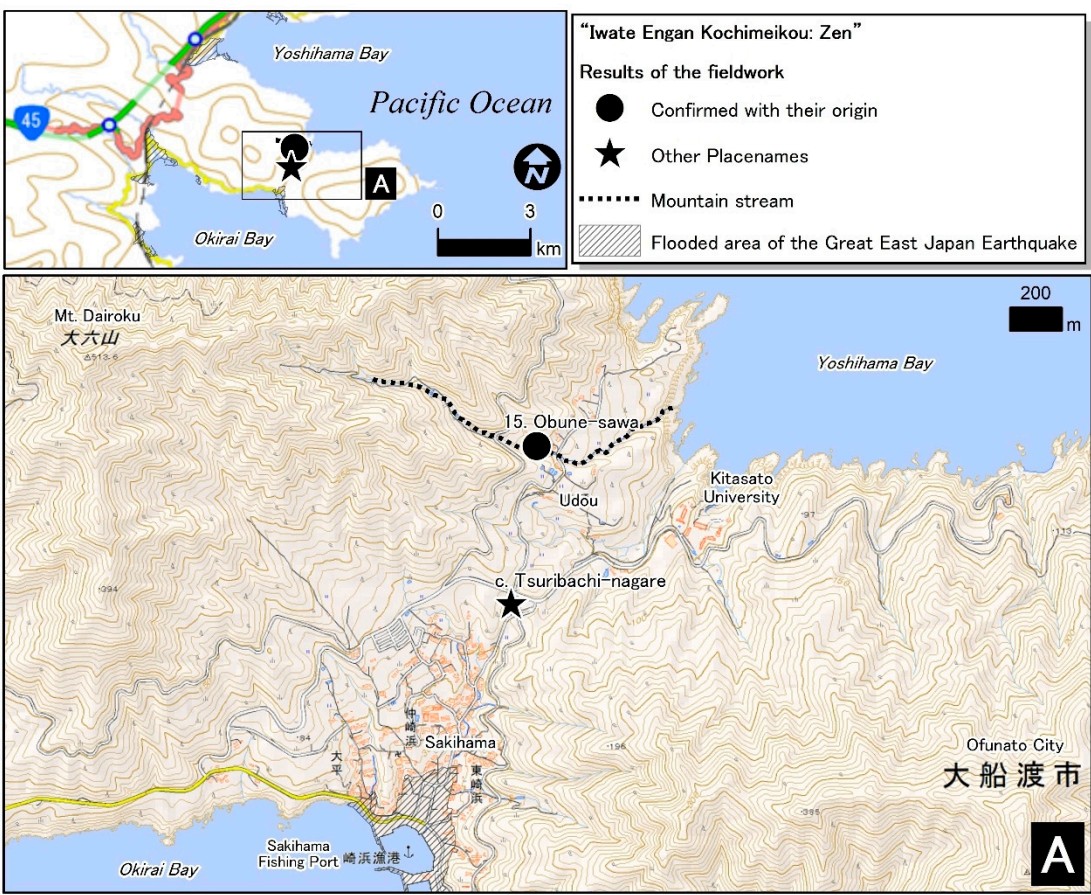

**Figure 3.** Map of Sakihama, Ofunato City. (Background: GSI Electronic Topographic map 25000).

Yamana's account [38] tells us that a large tsunami that crossed over a mountain pass between Udou and Sakihama had brought a large boat to a stream in the mountains, which was named Obune-zawa (Large Boat Stream). Prior to the group interview, we had found that at least a few villagers know about the location and the origin of Obune-zawa; and that the mountain pass is named Tsuribachi-nagare (Hanging Bucket Pass) after a tsunami delivered the hanging bucket, used on boats to pick seawater, to the pass (Figure 3); and that there is also a vague story about evacuation stating that villagers who evacuated in one way survived but those went the other way all died.

We invited the members of Sakihama Public Interest Body, the owner of the community forestry acting also as a neighborhood association, to the group interview session at Sakihama Public Hall, August 21, 2018. Eleven men in their 50s–70s appeared to the group interview.

All attendants knew Tsuribachi-nagare because the place is used for a meeting place, the name is used to refer to a place for communal work such as weeding, and because there had been a forest owned by local elementary school for education and attendants used to study there. We found that Obune-zawa refers not to the entire stream but a point where the stream meets the footpath to the Yoshihama, a neighboring village. Attendants were unanimously specific about the tsunami that crossed over Tsuribachi-nagare, which came from the other side of the pass from the northeast. Udou, the other side of the pass, was not inhabited until the post-WWII settlement. The villagers from Sakihama do not live there because it is colder there because of the northerly wind.

On the story of evacuation, attendants told that villagers who evacuated to the mountain in the west all died but those who evacuated to the mountain in the east survived. Attendants discussed that villagers survived because the mountain in the east is steeper so the tsunami could not reach there, or villagers died because the tsunami wave from the northeast went over the pass and to the mountain in the west.

There were no occasions specifically to learn the history and folklore of the community, and origins and stories are told by fathers and grandfathers and discussed during occasions such as communal work or when practicing *ken-mai*, a local sword dance. Attendants reckon that generations younger than those in their 40s do not know the placenames and the origins, as the number of communal work decreased and the placenames are used less often, and because the younger generations do not join the gatherings. Several attendants remembered and sang the tsunami song learned in school, but the song is no longer taught after the local elementary school is merged to the nearby school. Fewer attendants knew about Obune-zawa because the path is rarely used since a road has been built.

Attendants believe the origins and the stories are true. In response to our doubt that the pass at an altitude of 80 m may be too high for a tsunami to cross, attendants explained that the land is rising and thus the pass was lower and the sea was higher when the event occurred. They believe so because shellfish fossils are found on mountain tops and relics of the Jomon Period (circa 14,000–300 BCE) is found at altitudes of 100 m. An attendant told us that when he encountered the Great East Japan Earthquake, he immediately remembered the origins and felt frozen, yet he did not evacuate as he was already at a higher ground.

The attendants advised us to meet an elder who is most knowledgeable of the history of the village, so we did on the following day. He told us about seaweed got tangled on the moth tree of the village temple at a high ground, which we measured that it was at the altitude of about 30 m. He also said that the story of the crossing tsunami and that of the evacuation is not necessarily of the same disaster.

### 3.2.2. Unosumai, Kamaishi City

Unosumai consists of settlements along the Unosumai River (Figure 4) and is adjacent to Kamaishi City, the major city in the Sanriku Coast. The town is within the administrative boundary of Kamaishi City since 1955 and the town center at the mouth of Unosumai River close to the sea became more urbanized. Tsunami 2011 completely destroyed the town center. The death toll in Unosumai Town was 580 persons out of a population of 6630.

The town is the site of the miracle and the tragedy of Unosumai of the Great East Japan Earthquake and Tsunami 2011. The miracle refers to the evacuation of the students of Unosumai Elementary School and Kamaishi Higashi Junior High School. The two schools situated next to each other were not within the predicted tsunami inundation zone, but they have been trained of hazard mitigation by Professor Toshitaka Katada of Gunma University. The junior high school students started to evacuate to the evacuation site they have previously decided at the training, and the elementary school pupils followed suit. They even brought kindergarten pupils they encountered along the way to the evacuation site. They, after seeing the cliff behind their evacuation site is collapsing, decided to move to the evacuation area designated by the City. Then they saw a gigantic tsunami approaching so they evacuated even further to the higher ground along the national road. The tsunami reached near the

designated evacuation area but all the students saved their lives [49]. The miracle was applauded as the students have put *tsunami-tendenko* into practice.

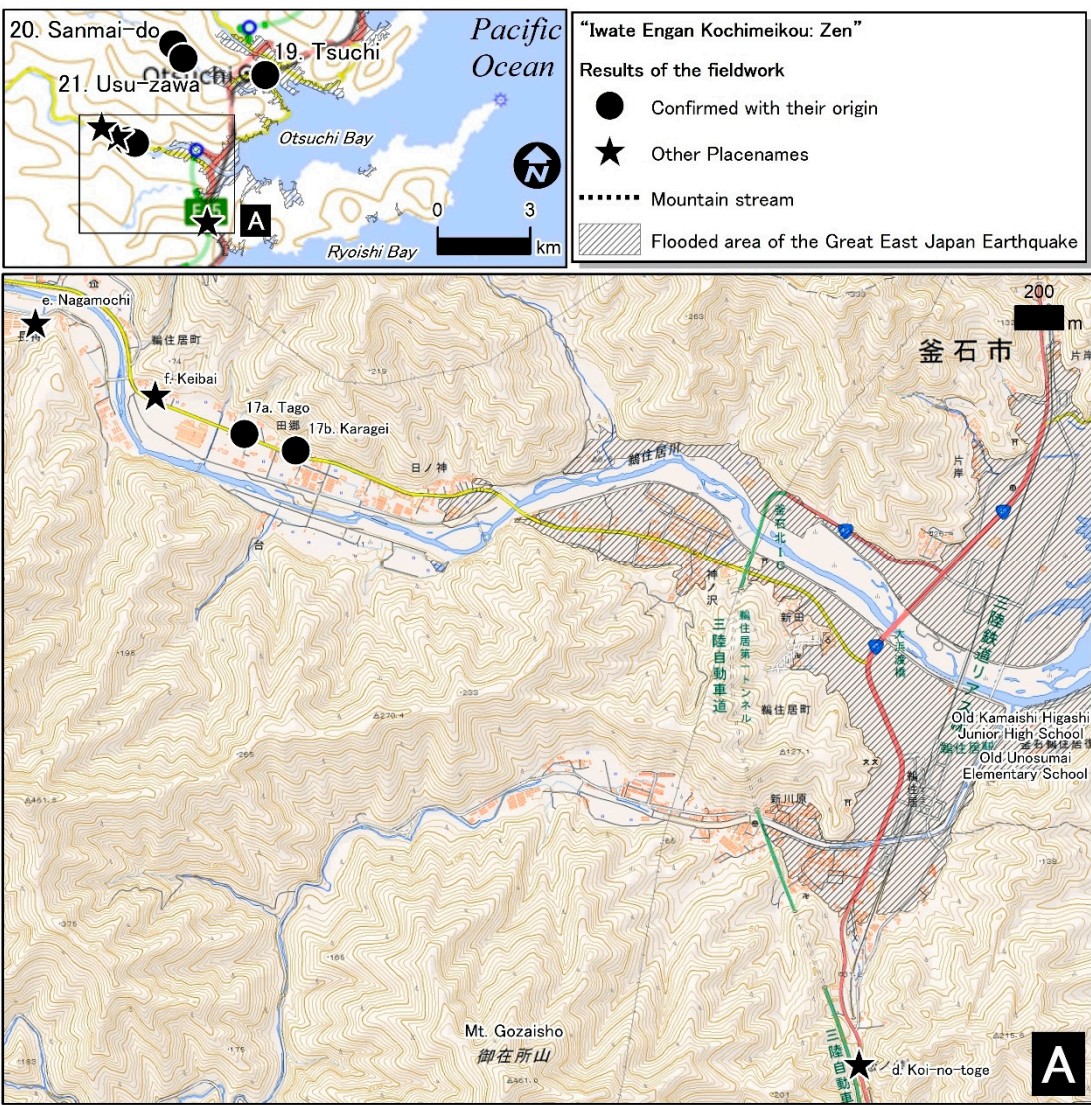

**Figure 4.** Map of Unosumai, Kamaishi City. (Background: GSI Electronic Topographic map 25000).

The tragedy refers to large number of death at Unosumai Disaster Mitigation Center. Although this facility was not a city designated evacuation site, the local neighborhood association had been doing evacuation drills at this site because the city designated evacuation site was far away and the attendance rate for training was low. Mislead by the name of the center and the previous drills, many evacuated to this place and only 34 survived. The number of evacuees to this site and thus the total death toll is unknown. Kamaishi City reckons that the number of evacuees is 100 whereas the Association of the Bereaved says the number is 248 [50].

Yamana's list [38] have Tako-karakai as a tsunami-originated placename for this area. Prior to the group interview, we had learned that the name of a settlement Tagou comes from *tako*, an octopus, being delivered to the area. We did not know what 'karakai' refers to, but there was a house in Tagou settlement whose name was Karagei and that *karagei* is a local name for a small ray fish. We further learned that there are two more settlements along the river having names of objects delivered from a tsunami, Keibai (a type of olive snail) and Nagamochi (a portable wardrobe). Unosumai also have a legend that a large tsunami crossed over Koi-no-toge, which have given the Chinese characters to

mean the mountain pass (*toge*) of love (*koi*), but it is interpreted that the ominous origin of tsunami 'crossing over' (*koe*) is intentionally masked by assigning a Chinese character for love (*koi*).

We have invited residents of Unosumai Town for the group interview session at Unosumai Public Hall on August 23, 2018, but only two appeared, a retired civil servant of Kamaishi City, and a member of a group studying placenames founded after the Great East Japan Earthquake to restore the traditions and memories of the much-affected region.

They told us that the origins of Karagei, Tagou, Keibai, and Nagamochi were forgotten until the Great East Japan Earthquake 2011. Being close to a large city, the number of houses in Unosumai have increased at the town center. New residents did not know the placenames or believed that the tsunami would reach the area. An attendant in his 70s says everybody in his generation heard the legend from elders when they were little but doubted after they have grown up. Both say they did not tell these stories to their children.

When we asked whether they believe the origin of the placenames, they univocally said they "believe 100%" even though the Tsunami 2011 did not reach Karagei, the lowermost of the four. They proposed to us to make a map of tsunami-originated placenames and house names. They advised us that a map-making should be carefully based on historical sources and said, "because people may tell you that you are liars". An attendant thinks tsunami ruins, which the City plans to demolish, should be kept for commemoration because he thinks the memory will be lost soon, reckoning that stories can last for only 20–30 years.

We also interviewed an old man in his 80s from the region, who is working as a volunteer guide to tourists to the area and telling the origins of Tagou and Nagamochi. He is a modern Cassandra (a Trojan prophetess whose accurate oracles are cursed to be believed by nobody, in Greek mythology). He laments, that children of Unosumai were good, but adults were 'complete nonsense'. He has been warning of tsunamis with the origins of Tagou and Nagamochi, but people disbelieved. He recalls a local welfare committee member commented, "you are so good at telling tall-tales"; a man from the region who died of the Tsunami 2011 was telling him "tsunami would never reach here". He continues, "When I was asked to do a lecture on local stories, I asked the participants whether they know the elevation of the town center. They did not even have the knowledge that it was less than 3 m, but still, they tell me that I am a liar". With anguish, he says, "why would I tell a lie that does not benefit me?"

Despite the Tsunami 2011 destroying most of the town center and killing many lives, he did not earn his redemption and kept his Cassandra-like character; when we were searching for his contacts to interview him, people advised us not to believe him at face value because he is a tall-tale-teller. That taught us that not only the new residents but also the people from the region are also refusing to hear his stories.

He told us that the four placenames are all originally house names of representative family in each settlement and that the house Nagamochi of the settlement at the most upstream had previously located several hundred meters downstream, at a place where it used to be the edge of a marshland for retarding flood; the house relocated because of frequent flooding. We confirmed marshland-like feature and a shift in settlement name "Nagamochi" in the 1953 edition of GSI 1/50,000 Topographic Map. The retarding marshland is now reclaimed and converted into paddy fields. He argued that if marshland were not reclaimed and if river dikes were not in place, the Tsunami 2011 may have reached Nagamochi at its original place.

He says the local history and legends are not discussed among local residents nor are they taught in schools. The stories are only told within the family, and he learned them from his grandfather because his father was busy at work. He suggests that passing on the local knowledge would be difficult in a nuclear family which has become the standard.

### 3.2.3. In Other Communities

For the placenames indicating extremely high tsunami run-ups, the fieldwork conducted between in 2015–2016 for the previous study [39] had asked in some detail on the perception the tsunami-originated placenames, and will be used here.

In Tarou, Miyako City, there is a settlement named Koeta on top of coastal terrace of about 100 m, of which its origin being that a tsunami *koeta*, or 'crossed-over'. The origin is not obvious from the Chinese characters given to the phonetic name. As we asked the origin to the local residents, people said that it is because the settlement is located over (*koe*) the mountain from the Tarou center, but as we disclosed our research purpose of seeking the tsunami-originated placenames, they told us that there is a legend that tsunami has crossed over that place. They also taught us that the nearby mountain ridge has the name Matsu-nagane, with the origin saying that tsunami have delivered pine (*matsu*) to the ridge (*nagane*), or that tsunami washed away everything but pine trees. People believed that the origin is a true historical event, but that it would not happen today because the land is rising and the mountain and the ridges are higher now. Accordingly, nobody evacuated during the Tsunami 2011, also because there were nowhere higher than that settlement in the vicinity.

In Iwaizumi Town, there is a settlement called Moshi attached to a steep slope facing the sea, in the elevation between 30 to 80 m. Behind the settlement on the top of a coastal terrace exceeding 120 m in altitude, there is an agricultural field named Hachi-mori, with the origin being a bowl (*hachi*) arriving after a tsunami. According to the residents, there are also fields named Bozu-bata and Wano, with the origins that a monk (*bozu*) and a small bowl (*wan*) were delivered by a tsunami. Again, people believed the origins to be true but did not evacuate during the Tsunami 2011 with the same reasoning with Koeta and in Sakihama, that tsunami would not reach that height in the present day. At the foot of the settlement at a very visible place, there is a stone monument commemorating the Showa Tsunami 1933 stating that there would be a tsunami after a large earthquake, and to evacuate to the elevation or higher to that of the monument.

## 4. Discussion

The examination of the reality of the 21 tsunami-originated placenames revealed that a significant proportion of the placenames had origins indicating unrealistic tsunami reach. The probability that Tsunami 2011 would reach the altitude and the distance by a tsunami-originated placename was less than 0.1% for nine places, seven of which exceeded the altitudes of 80 m and the probabilities were less than 0.001%. We evaluated that the origins of these placenames are not factual truth despite comparison has been made only to one specific tsunami, because the Tsunami 2011 is considered to be one of the largest in the last 1500 years [41]. The highest tsunami run-up ever recorded in Japan used to be 85 m for the Meiwa Tsunami 1771 in Ishigaki Island, Okinawa Prefecture, which also had a legend of the waves crossing over a mountain pass. A thorough geological examination and a careful review of the historical documents concluded that the primitive survey method overly stated the height and the actual height should have been around 30 m [51]. These suggest that exaggeration in disaster experience in the placename origins are very common.

On the other hand, there were 10 placenames indicating tsunami reach within 99% prediction interval of the Tsunami 2011, all of which were less than 30 m in altitudes. We evaluated these as realistic despite only two of them were actually inundated by the Tsunami 2011, because there were several drainage basins that Tsunami 2011 exceeded this threshold (we had 1,005 basins), the past tsunami can be somewhat greater than Tsunami 2011, and the modern seawalls, breakwaters, and other constructs could have reduced the tsunami reach. Ebina finds that if the past topography is restored and artificial structures such as breakwaters and river dikes are removed, tsunami similar to Tsunami 2011 can reach to places where statements of tsunami attack in the historical documents have been considered false [52].

The result showing that about half of placenames indicating unrealistic tsunami reach suggest that verification of each observation is critical [5,22,24], if the research were to retrieve past tsunami reach

from the placenames. We omitted placenames that have large spatial extent from the analysis, like the mountain and the entire stream, because a location of tsunami reach cannot be identified. A placename from a house name has to be taken with care because the location of the placename can move together with the house, especially in a tsunami-prone area where relocation after a tsunami attack is usual. The placenames indicating unrealistic tsunami reach (as well as the more realistic ones) tended to cluster spatially. If such cluster were created out of spatial dependence such that an exaggerated legend propagating to neighboring areas or that a place is named to be more consistent to the other, then spatial analyses used to remedy the errors in other cases [14,15] would not help. A cluster of high values for tsunami reach would not suggest the existence of such tsunami, they might be errors altogether. However, a cluster of tsunami-originated placenames does suggest the existence of some kind of disaster subculture, and can also be a reason why the origins have survived in such area.

Our objective of assessing the reality of placename origins was to compare the perceptions of the tsunami-originated placenames between the more realistic and the less realistic ones, to see how the placenames act as disaster subculture, and to obtain implications for disaster risk communication. Against our initial premise that the origin of a placename has to be true for it to function as disaster subculture, Unosumai having more realistic tsunami-originated placenames were dismissing the tsunami origins, whereas Sakihama and two other areas having unrealistic tsunami-origin were believing the origin as factual truth. Our initial premise needs to be thought out thoroughly. What is 'true' in everyday life may need to be distinguished from scientific and factual truth. Kamaishi City reports an account from a person from Unosumai after Tsunami 2011: "I realized that the legend of Tagou (octopus) and Karagei (small ray fish) [being delivered by a tsunami] was not a lie!" ([53], p. 261). The tsunami did not actually reach those places, but for a person who did not believe of a tsunami that would completely destroy the town center, the legend was more 'true' than what he had believed.

People in Unosumai, with a series of realistic tsunami-originated placenames, have tried to discredit the origins passed on to families in the area. We had just one case but since it is rare to witness a tradition just about to disappear, some lesson should be learned. Realistic origins contradict to the desire for safety and become subject to doubt. People have heard of the legends but tried to deny the inconvenient truth. People had discredited a man who tried to remind the origins, by stigmatizing him a liar and a tall-tale-teller, and that continued even after the Tsunami 2011 destroyed the town center. This can be regarded as a part of normalcy bias.

Belief/disbelief is also affected by whom the information is given. The community maintained the inconvenient origins when they were told from their own forebears. As the area became urbanized and the new residents were told of the stories from indigenous residents, doubts started to creep in. By the time the new residents outnumbered the indigenous residents, the origins and legends are mythified, including among the members of the families living there for many generations.

Different perceptions across age also point that contradiction to the desire for safety leads to dismissal of an inconvenient truth. People believed the origins when they were told at childhood but doubted as they grow up. Increased responsibility to one's family and assets created repulsion to the inconvenient truth, people opted to dismiss the inconvenient truth. The last significant tsunami damage was from Chilean Tsunami 1960, that was a long time ago and much smaller than what the tsunami-originated placenames imply. The tremendous threat the tsunami-originated placename suggest was too stressful to live by during the long normal period of the low-frequency high-risk hazards. So, 580 adults died from the Tsunami 2011 [49]. On the other hand, children who learned *tsunami-tendenko* tradition, an evacuation first practice, at schools, evacuated voluntarily and accordingly to the felt magnitude of the earthquake and the witnessed height of the Tsunami 2011. No children at schools (except 5 at home) were killed [49].

In Sakihama and in the other two communities, having places with tsunami origin exceeding the height of 80 m, the implication of the placename did not contradict to the people's desire for safety. Tsunami height indicated by the placename origins were too high to believe it will happen in the same way, and people devised a way to understand the origin. People say the origins are factual

truth, but with a surprisingly unanimous explanation in three different communities: The mountains used to be lower. The placename, shell fossil and ancient relics found on higher grounds, and other tsunami-originated placenames were all consistent and made sense (all three communities had multiple exaggerated tsunami-originated placenames). The placename origins survived because they were embedded in the system of indigenous knowledge that consistently explains the intriguing facts in the area. For scientists, as we have seen, Tsunami 2011 proved the origins of those unrealistic tsunami-originated placenames are false, with the statistical significance of 0.001%. For people in those communities with their systems of indigenous knowledge, the same tsunami 2011 proved the origins are true, reaching to the height that might have been at the time mountain was lower. After all, the exaggerated tsunami-originated placenames were accurate in predicting the arrival of apocalyptic tsunami to the livelihood sphere of the communities.

People do not believe that tsunami will reach the same height today, because mountains are higher now. Accordingly, nobody said he or she evacuated during the Tsunami 2011. Then in which ways would the tsunami-originated placename act as disaster subculture? The ambiguity in the exaggerated tsunami origins created tension that nowhere would be perfectly safe (or unsafe) and kept people alert to tsunamis. Although lower-ground is more convenient to go to the fishing ports, a majority of residents in Sakihama and all residents in Koeta and Moshi had lived at higher-grounds where Tsunami 2011 did not reach. We failed to ask whether there was any preference for housing site before the Tsunami 2011, but we encountered several individuals who moved his house from the lowland to higher ground for safety before the Tsunami 2011 when the circumstances allowed to do so.

The strength of exaggerated tsunami-originated placename came from its ambiguity, arising from its unrealistic height requiring an argument of changing mountain height. Its accuracy in predicting the arrival of an unimaginable tsunami, enabled it to act as disaster subculture to transmit the awareness to low-frequency high-risk hazards. On the other hand, the precision of the realistic tsunami-originated placename indicating the exact location of a tsunami reach was the weakness. Its contradiction to people's desire for safety made people doubt its trustfulness and were repeatedly refuted during the long normal period and by the smaller tsunamis not reaching the place.

Such difficulty in risk communication for low-frequency high-risk hazards would also apply to modern scientific hazard information. The spatial extent under risk is large and usually occupied by a large number of people having the desire for safety. Long normal period gives ample time not only for oblivion but to refute the information. Combatting normalcy bias and the maintenance of general knowledge on high-risk hazards, through exploration of historical disasters and dissemination of large-scale disasters elsewhere, are the priorities of risk communication for low-frequency high-risk hazards.

Future prospects for tsunami-originated placenames as a medium [32] to maintain awareness to low-frequency high-risk hazard are dim, however. The placenames were used and origins transmitted while doing communal work, at social gatherings or at pastimes with the neighbors. All of these are in decline with modernization and depopulation in rural areas.

This paper has combined GIS and statistical analyses of tsunami-originated placenames with in-depth interviews on the perceptions of the origins. It demonstrated that disaster-originated placenames offer materials to understand how local people, in everyday life, understand and act upon natural disasters [36,37]. We highlighted the complexities of risk perceptions by comparing the communities with unrealistic and more realistic disaster-originated placenames.

Our discussion and implications remain speculative, however, because of the small number of communities we examined. There can be many other causes that a community stops in believing the local knowledge. There was some other disaster subculture(s) in place in the communities continuing to believe the placename origins, such as the stone monuments, the tsunami song, and *tsunami-tendenko* tradition, as well as the modern hazard mitigation knowledge for the area. Evaluating the relative importance of these factors in determining the belief/disbelief of the placename origins requires large number of samples that are not readily available, because of lack of comprehensive list of

tsunami-originated placenames, and of the need to find willing volunteers for the interviews. That is why a case study like this is important and needs to be accumulated.

**Author Contributions:** Conceptualization, Yuzuru Isoda, Akio Muranaka and Go Tanibata; formal analysis, Yuzuru Isoda; funding acquisition, Akio Muranaka; investigation, Yuzuru Isoda, Akio Muranaka, Go Tanibata, Kazumasa Hanaoka, Junzo Ohmura and Akihiro Tsukamoto; methodology, Yuzuru Isoda, Akio Muranaka, Go Tanibata, Kazumasa Hanaoka, Junzo Ohmura and Akihiro Tsukamoto; visualization, Akio Muranaka; writing—original draft, Yuzuru Isoda; writing—review & editing, Yuzuru Isoda, Akio Muranaka and Go Tanibata.

**Funding:** This work was supported by the Science Research Promotion Fund of the Promotion and Mutual Aid Corporation for Private Schools of Japan.

**Acknowledgments:** We would like to thank Yoshinori Kariya, Shigetoshi Fujiwara, and Yukio Iwasaki for kindly organizing the group interview sessions, and the residents of Sakihama and Unosumai for sharing their local knowledge and their views. We would also like to thank Shoichi and Ryoko Maekawa for providing the lead information at the initial phase of this study. Kohei Mori and Yui Itaya provided assistance during and after the fieldwork. We would also like to thank the three anonymous reviewers for their help in improving the paper.

**Conflicts of Interest:** The authors declare no conflict of interest. The funders had no role in the design of the study; in the collection, analyses, or interpretation of data; in the writing of the manuscript, or in the decision to publish the results.

## Appendix A

**Table A1.** Tsunami inundation and run-up values for Figure 1. The *t* values and the corresponding probability that Tsunami 2011 reaches the altitude and distance indicated by the placename is based on an empirical relationship in Appendix B.

| ID | Placename | Tsunami Inundation and Run-Up | | | | | | |
|----|-----------|----------------------|----------------|----------------------------|----------------|---------|-------------|---|
| | | Placename | | Haraguchi & Iwamatsu [44] | | TTJS Data [45] | | |
| | | Altitude (m) | Distance (m) | Altitude (m) | Distance (m) | *t* value | Probability | |
| 1 | Same-ga-fuchi | 13 | 1614 | 12.0 | 1419 | 0.765 | 0.222 | |
| 12 | Kuwa-dai | - | - | - | - | - | - | |
| 15 | Obune-zawa | 100 | 640 | 10.0 | 55 | 4.727 | 0.000 | *** |
| 17a | Tagou | 14 | 4312 | 8.4 | 3872 | 1.212 | 0.113 | |
| 17b | Karagei | 12 | 4069 | 8.4 | 3872 | 0.876 | 0.191 | |
| 19 | Tsuchi | - | - | - | - | - | - | |
| 20 | Sanmai-do | 9.7 | 3977 | 15.0 | 4119 | 0.430 | 0.334 | |
| 21 | Usu-zawa | 9.2 | 3355 | 15.0 | 4119 | 0.270 | 0.394 | |
| 22 | Kujira-yama | - | - | - | - | - | - | |
| 33 | Koe-ta | 104 | 1336 | 35.0 | 485 | 5.006 | 0.000 | *** |
| 34 | Hachi-mori | 130 | 728 | 27.4 | 103 | 5.315 | 0.000 | *** |
| 40 | Mekka-zawa | 24 | 1037 | 13.5 | 50 | 1.898 | 0.029 | |
| a | Atta | 28 | 485 | 16.2 | 447 | 1.993 | 0.023 | |
| b | Mizu-ai-toge | 35 | 601 | 24.3 | 476 | 2.517 | 0.006 | ** |
| c | Tsuribachi-nagare | 84 | 946 | 20.0 | 263 | 4.465 | 0.000 | *** |
| d | Koi-no-toge | 45 | 1045 | 21.4 | 683 | 3.199 | 0.001 | *** |
| e | Nagamochi | 19 | 4966 | 8.4 | 3872 | 1.884 | 0.030 | |
| f | Keibai | 16 | 4736 | 8.4 | 3872 | 1.515 | 0.065 | |
| g | Aburakko-zawa | - | - | - | - | - | - | |
| h | Yagi-fune | 40 | 1845 | 15.8 | 968 | 3.125 | 0.001 | *** |
| i | Matsu-nagane | 94 | 955 | 27.1 | 592 | 4.702 | 0.000 | *** |
| j | Bozu-bata | 125 | 601 | 27.4 | 103 | 5.190 | 0.000 | *** |
| k | Wano | 125 | 658 | 27.4 | 103 | 5.209 | 0.000 | *** |
| l | Aburakko-gappa | 30 | 1448 | 11.6 | 1129 | 2.459 | 0.007 | ** |
| m | Tago-ori | 20 | 672 | 13.3 | 87 | 1.393 | 0.082 | |

**, *** are probabilities less than 1% and 0.1%, respectively.

**Appendix B**

An empirical relationship between the gradient of a river and the distance tsunami traveled from the sea in Kayane et al. [46] is applied to the locations of highest Tsunami 2011 inundation or run-up in each drainage basin in the coast of Iwate Prefecture found in the TTSJ data [45] (Figure A1). The equation is estimated for 1005 basins using OLS as follows (standard error in parentheses):

$$\log y_i = 2.851 - 0.873 \log x_i + \epsilon_i$$
$$(0.029) \quad (0.012)$$

The $R^2$ is 0.833 and standard error of the regression is 0.423. One-sided prediction interval for 99% and 99.9% are derived, and they are mapped into Figure 2.

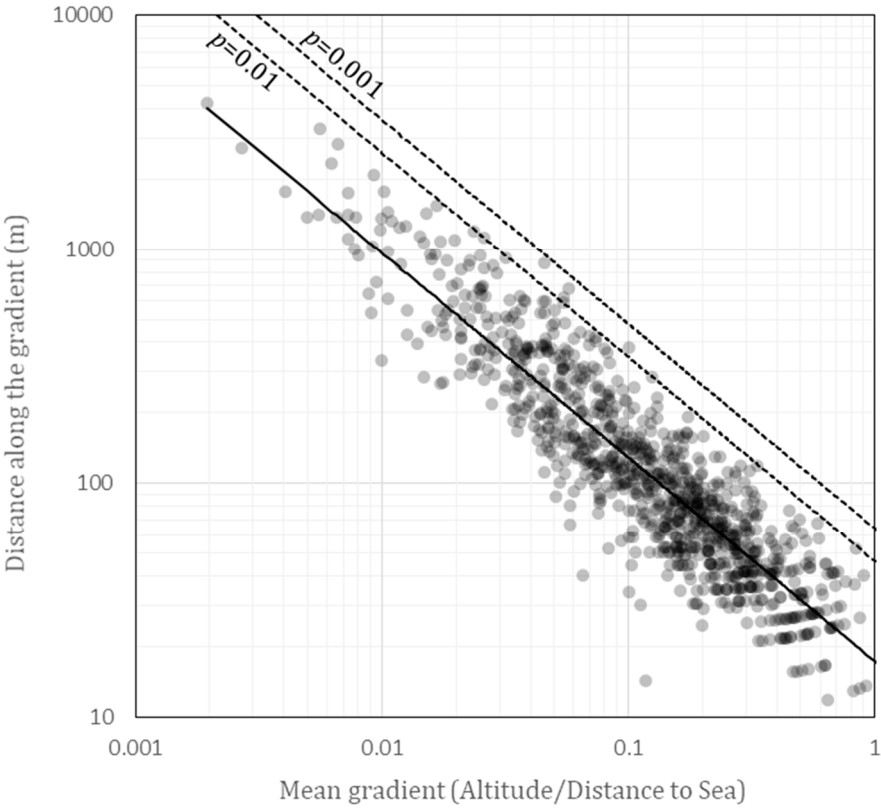

**Figure A1.** The empirical relationship between the mean gradient and the distance along the gradient for the highest Tsunami 2011 inundation and run-up for each drainage basin in the TTJS data [45]. The solid line is the mean and dashed lines are one-sided prediction interval of 99% and 99.9%.

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
