# Peer review of "Strengths of Exaggerated Tsunami-Originated Placenames: Disaster Subculture in Sanriku Coast, Japan"

_ijgi, doi:10.3390/ijgi8100429_

Round 1

Reviewer 1 Report

This is a solidly researched paper that covers a fascinating subset of place-names, and is methodologically sound. In terms of process and argument, I cannot find fault with the approach, and the claims of the authors are backed up and well-reasoned – despite the fuzziness of their results (I dislike the use of ‘probably’ without a solid statistical breakdown). The novel combination of folk etymology and scientific survey is done well for the most part. A very interesting read, on which they should all be commended. There are, however, a number of issues (one of which a major potential concern) which should be addressed before the piece sees publication.

-       The opening sentence, which claims that name studies ‘rarely resulted in acadmic [sic] findings’ – this is very much not the case from a linguistics or sociological perspective. The onomastic field has been booming in academic research and output (at least, in Western academia) since the 1950’s, and I believe in Eastern work since at least the 80’s. A revision of this section to acknowledge this is important, as it does alter the perceived impact of the paper.

-       There are a number of spelling and minor grammatical issues throughout the paper – for an example, see the quotation above, in which ‘academic’ is misspelled. A thorough read-through of the paper is required.

-       Several bold and generalised claims are made throughout the piece, such as ll. 56-57 and ll.140-141. Reference to supporting evidence would significantly strengthen such claims, and better situate the paper with existing work.

-       There are sections of unnecessary hyperbole and editorialising, such as ll.343-346. These would be best removed, and used instead in live presentations by the authors as supplemental side-details. Any content which has no bearing on the actual results should be considered cuttable.

-       Ethics statement – given the data collection method, did the authors receive ethical permission from their institutional body? Given the descriptive detail in the write-up and the acknowledged prominence of house-names (both shown especially in the previous point raised), identifiability is a major concern. Such an evidenced statement is essential for work of this nature – and I hope that all necessary consents were likewise gained.

Author Response

Thank you so much for the encouraging comment.
We revised the following points.

- On the fuzziness of the analysis,
We added another analysis for the assessment of the reality of tsunami-originated placenames.

- On the opening sentence,
We have rewritten to reflect what has been said in the literature, although we could not extend the coverage to onomastics in general.

- On spelling and grammatical issues,
We checked thoroughly.

- On bold and generalized claims,
We added references that were lacking.

- On unnecessary hyperbole
We did not remove the part about the continuation of stigmatization (ll 388-392), because our observation confirms the stigmatization, and that it was continuing. We revised to let readers understand that.
We also moved most of the interpretations in the result section.

- On Ethics statement
We did not take ethical permission from our institutions because nothing personal and confidential was expected. All consent is gained from the participants. We did find possible identifiability for a man in Unosumai, so we blurred his profile. The house-names are not confidential and there are no ethic concerns.

Sincerely

Reviewer 2 Report

My one comment is about the terminology of "reality", "true", "realistic", "wrong", etc. I believe a more nuanced use of this terms is warranted as you are assigning place names to specific points on a map, when they may actually represent larger areas, and therefore may not be "wrong".

Also, a limitation section is missing from the Discussion section.

Author Response

Thank you so much for a favorable comment.

We rethought on the use of terms "reality", "true", etc.
Our discussion has changed and developed significantly.
Thank you so much for the point you raised.

We added the limitations section in the very end.

Sincerely,

Reviewer 3 Report

In my opinion, reviewed paper has strong potential, because of its content and application possibilities! However, several changes should have been done. It´s important to more clarify aiming of the article. Is it verification of trustfulness of tsunami place names by GIS? Or is it research on local knowledge of these place names and its legacy? I would prefer first option, because in the second case, there is too small group of respondents for qualified results. Results from group interview should refine the results from GIS analysis and show limits of working with datasets of place names in GIS environment (example of house renaming and shifting of place names, or case of Obune-Zawa). It could enrich the discussion and show the limits.

Major comments:

·         Introduction: Authors used a term “thematic toponymy” is it their own, or they have support in literature? I think that, thematic works with place names are connected with onomastics research from the beginning. The term should be more discussed. From the focus of the article, it is important to underline connection of place names and GIS – for example the term “Toponymic GIS” (see Fuchs S. (2015): Toponymic GIS – Role and potential of place names in the context of geographic information systems and GIS. Kartographische Nachrichten, 2015(6): 330–337.)

·         Introduction: There is no explanation of the term “disaster culture” in the introduction, maybe some short definition support by references, could help

·         Methods: lines 239-240 should be part of methods – information’s about group of respondents. Why authors use group interview in this case, is there any similar research where it was used (some references)

·         Results: I don´t know if I can agree with the sentence (line 230-231) “We hesitate to totally deny the origins of these places because there is no scientific method to know the past tsunami run-ups” – If is it connected with place names - I think, that could be huge benefit because, they can be bearers of landscape memory, lasting several centuries, when there is no other scientific evidence

·         Results and discussion: I think that some parts of results should be shift or embedded to discussion part (lines 281-291, shorter version of text in lines 327–346, shorter version of chapter 3.23).  There should by some mention and comparison to results of other studies (with references) in the discussion. It concerns both the methodical and the result part. For example for the importance of verification of the mapping and GIS results with local knowledge, local actors or practitioners (see Penko Seidl (2008):  Significance of Toponyms, With emphasis on field names, for studying cultural landscape, Acta Geographica Slovenica, 48 (1), 33-56. Wang, F et al. (2014): Mapping and spatial analysis of multiethnic toponyms in Yunnan, China. Cartography and Geographic Information Science, 41(1): 86–99; Frajer, Šimáček (2019): Localisation of the painter’s canvas: landscape paintings from the Iron Mountains (Czech Republic), https://doi.org/10.1080/17445647.2018.1563570)

Minor comments:

·         Line 44 – citation with Frajer and Feidor – according to reference list correct is: Frajer and Fiedor

·         Table 1 and Line 187 correct spelling is Aburakko-sawa or zawa?

·         Appendix A -  May be adding a column with difference between GIS data and Tsunami 2011 data, should provide quick information

Author Response

Thank you for thoroughly reviewing the manuscript and helpful references.

We revised the following points in response to your comments

- On the aiming of the paper,
We have rewritten the research purpose section (ll.112-120), but we maintained that our purpose is on understanding the perceptions of disaster-originated placenames and their roles in disaster mitigation.

- On the term "thematic toponymy"
We are continuing to use the term but removed from the title because "thematic toponymy" is what we observe as a trend in toponymic studies and not an approach that we are proposing. Your references let us find many Critical Toponymic studies using GIS and spatial analysis, and was added (ll. 52-55).

- On the term "disaster culture"
We added references to the definition of "disaster (sub)culture". As we checked for recent literature on disaster culture, we found that more researchers are using the term in its original form as coined in the 1960s, so we followed that and change all the previous "disaster culture" to "disaster subculture".

- On the group interview method
We added details on our method for the group interview (ll. 210-217)

- On results,
We think we have the same sentiment about disaster-originated placename as you do, and have no intention to say they are useless if they do not indicate the tsunami run-ups. We removed the part you mentioned to avoid misunderstandings.

- On results and discussions
We embedded our results into discussion as you advised us. At the same time, we kept interpretations in the result section to a minimum and shifted to the discussion section.

- On the importance of verification
We added a section on how verification is done in toponymic studies (ll. 61-65), and also the implication from this issue from our study in the discussion section (ll 456-467).

- We corrected the mistake in the author name.

- We corrected the inconsistencies in the placenames

- On differences between the placename and Tsunami 2011.
As a reply to your suggestion, we added another analysis that derives the probability that Tsunami 2011 would reach to the altitudes and the distance the placename indicates. Thank you for this suggestion.

Sincerely,

Round 2
